# A generic method for PLC control system design based on Digital Twin

Hao Shi*, Qingliang Zhu, Jingde Bo, Jiayang Shi, Tao Zhao

Department of Electrical Engineering and Automation, Shanxi Institute of Technology, Yangquan, Shanxi, China

* 93051139@qq.com

## Abstract

This study explores a design method for electrical control systems based on the Digital Twin (DT) from the perspective of control engineers. The proposed method enables engineers to systematically develop and validate such systems, thereby enhancing the quality and efficacy of industrial digitalization. By identifying critical challenges in DT-based control system design, we proposed a maturity evaluation framework to ensure that the DT's fidelity aligns with the requirements of control system design application scenarios. Subsequently, a comparative analysis of virtual and physical commissioning was conducted, leading to the formulation of a structured implementation framework for DT-based control system design. A six-floor elevator DT was employed as the controlled object to demonstrate the proposed method. A virtual commissioning platform was constructed to validate the feasibility and reliability of the control system. This approach provides actionable insights for the corresponding physical entity to optimize construction schemes, commissioning workflows, and operation adjustments. The practical results demonstrate that the proposed DT-based design method of the electrical control system enables high-fidelity validation of production line design, reduces resource consumption during physical equipment commissioning, and significantly enhances development efficiency and quality while shortening construction and commissioning timelines.

## 1. Introduction

The rapid development of computer technologies, the Internet of Things, DT, and other technologies, coupled with the increasing demand for personalized and customized production in modern manufacturing [1], necessitates that next-generation manufacturing systems possess digitization, flexibility, agility, customization, networking, and globalization. These features enable rapid responses to customer demands and enhance competitiveness in global manufacturing. Digital manufacturing systems, urgently required by enterprises, serve as critical enablers for the digital

**Data availability statement:** All relevant data are within the paper and its Supporting Information files.

**Funding:** This work was supported by Shanxi Province online and offline hybrid first-class courses (Project No. K2022293); Shanxi Province Innovation and Entrepreneurship Training Project for College Students (Project No. 202414527009); Shanxi Institute of Technology scientific research project (Project No. 2023005); Shanxi Institute of Technology professional-innovation-entrepreneurship integration construction project (Project No. ZCRH014). The funders had no role in study design, data collection and analysis, decision to publish, or preparation of the manuscript.

**Competing interests:** The authors have declared that no competing interests exist.

transformation of manufacturing enterprises and the evolution of next-generation manufacturing systems [2].

As China accelerates its digital transformation in manufacturing, an increasing number of equipment and production lines are adopting DT models for virtual commissioning to validate feasibility and reliability prior to physical deployment [3]. This approach can reduce material waste, minimize costs from physical commissioning, and reduce the workloads of research and development. Iterative commissioning under varied conditions refines equipment and processes, yielding optimal solutions and rational parameter configurations, thereby shortening product commissioning cycles. These processes can also assess the DT completeness, improving model quality for subsequent production applications [4–6].

A comprehensive DT extends beyond mere replication of the production equipment. The mature DT should incorporate the DT of the electrical control system to enable precise control over the DT. During the design, commissioning, and process improvement of the production line, the DT control system can be used to operate the DT to identify potential issues. Solutions can be subsequently validated on the virtual production line to assess their effectiveness. Following successful virtual validation, optimized solutions are deployed on the physical entity equipment. This approach significantly improves the efficiency of production line design, commissioning, and process improvement while mitigating operational risks and reducing commissioning costs. As evidenced by this workflow, the DT of the electrical control system serves as an indispensable component of the complete DT for the production line or equipment.

In summary, as comprehensive participants in the entire lifecycle of manufacturing production line design, installation, commissioning, runtime, and maintenance, electrical control engineers will increasingly engage with digitized controlled objects. Mastering the design methods of DT-based electrical control systems will be a fundamental requirement for electrical control engineers. From the perspective of electrical control engineers, this paper explores the design methods of electrical control systems, using an elevator DT as a case study. Firstly, critical challenges in the design of DT-based electrical control systems are identified, and a structured workflow with its requisite commissioning environment is proposed. Next, the maturity metrics that DT should fulfill to support the application scenarios of designing electrical control systems are analyzed. Finally, through the implementation of designing the electrical control system based on elevator DT, we validate the feasibility of this approach. The proposed method and findings provide a referenceable pathway for electrical control engineers undertaking analogous projects.

## 2. Analysis of critical challenges in the DT-based electrical control system design

The concept and technologies of DT are continuously evolving and being refined, with varying interpretations and requirements across different industries and application domains. This study defines the DT from the perspective of electrical control engineers. A DT is a high-fidelity digital model that mirrors its corresponding physical

entity in mechanical, dynamic, and electrical characteristics. It enables the simulation of the corresponding physical entity's operational processes and dynamic states in virtual environments, generating data to diagnose and predict potential issues that may arise in the physical entity under real-world application scenarios. Furthermore, virtual commissioning via DT overcomes the constraints (time, space, cost, and safety) inherent to physical entities, validating the feasibility and reliability of the designed control system. The virtual commissioning results can further inform the construction and operation of physical entities, establishing a foundation for optimizing construction schemes and refining operation protocols [7–9]. The subsequent sections systematically examine critical challenges in control system design and commissioning based on DT.

## 2.1 Quality of DT model

The design and commissioning of control systems are fundamentally grounded in their control objects. Although the controlled object here is DT, the objective of designing DT-based control systems is to validate the design scheme of the corresponding physical entity and improve the efficiency and quality of the physical control system design. Additionally, the control system of DT should be capable of seamless migration to the corresponding physical entity. Consequently, the fidelity of the DT is a critical determinant of control system design, necessitating rigorous alignment between the DT and its corresponding physical entity in physical architecture, operational principles, and environmental conditions. Only control systems designed based on such high-fidelity DTs can effectively guide the construction and production process of their corresponding physical entities while enhancing the decision-making for the operation and optimization of these physical entities [10,11].

Although the DT is a digital model of a physical entity, it cannot completely reproduce physical entities in software. Overemphasizing digital model fidelity to the physical entity may significantly escalate the model complexity, thereby degrading critical performance metrics such as reliability, computational tractability, maintainability, and other essential characteristics [6,12]. Currently, the industry has not yet established a unified evaluation framework to assess the quality of DTs, thereby leading to extensive research efforts by industry experts and researchers. Zhang Lin et al. [13] proposed that the trustworthiness of DT models should be evaluated according to the specific simulation objectives and requirements. A model validated for one simulation scenario may fail to maintain its trustworthiness in another, as the same model may demonstrate significant variability in trustworthiness across different simulation scenarios. Building on their five-dimension DT model, Tao Fei et al. [2,14,15] categorized DTs into six maturity levels according to functions and application scenarios. This classification provides a systematic method to evaluate whether the existing DT meets predefined operational requirements and objectives.

In summary, within the application scenario of DT-based control system design, the quality of DTs can be evaluated by assessing the maturity levels across three critical dimensions: DT Model, DT Data, and Functional Service. This evaluation determines whether the DT meets the requirements of application scenarios.

## 2.2 The similarities and differences between virtual and physical commissioning

The design of DT-based control systems also requires iterative commissioning to determine the feasibility and reliability of the control system while establishing its system architecture and control logic. Furthermore, by analyzing the multi-dimensional data generated during the commissioning process, the control system can be progressively optimized until it meets predefined design specifications. The commissioning process based on DT falls under the category of virtual commissioning. Virtual commissioning requires establishing bidirectional communication between the virtual simulation model and the control system. The control system drives the operation of the virtual simulation model, while the virtual simulation model transmits real-time operational state data back to the control system for closed-loop validation [8,16].

The relationship between virtual and physical commissioning is shown in Fig 1. Although the core procedures and methods of virtual commissioning are similar to those of physical commissioning, the absence of a physical

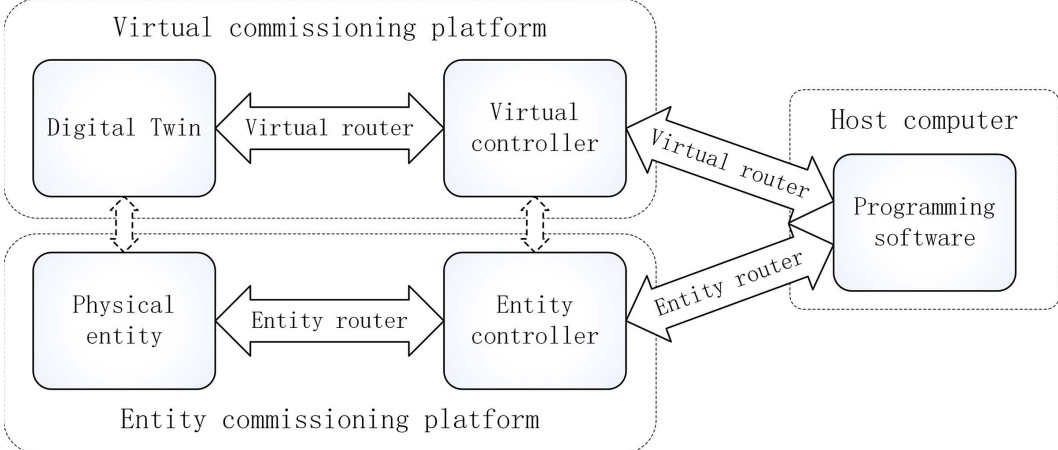

**Fig 1. Relationship diagram of virtual and physical commissioning.**

entity necessitates the prior establishment of a virtual platform to achieve bidirectional communication between the control system and the DT, enabling real-time interaction. Therefore, the primary task of virtual commissioning is to build a virtual commissioning environment that ensures the real-time interaction between the control system and the DT, as well as the visualization and analyzability of the test data. The construction of the virtual commissioning environment mainly includes two key components: the selection and configuration of controller-related hardware and software and the selection and configuration of communication protocols and interfaces for real-time data transmission.

The controller, as the core component of the control system, should first be determined when constructing the virtual commissioning environment. The controller may be selected as either the physical controller or the virtual controller. The controlled object of virtual commissioning is the DT, which is a virtual model based on a computer system. Consequently, with advancements in computational power and the maturation of virtual controller technologies, the selection of virtual controllers has increasingly become a more convenient approach in virtual commissioning [17–20]. Subsequent to controller selection, a communication interface must be established between the controller and the DT. If the virtual controller is selected, real-time communication between the virtual controller and the DT is typically achieved through software. This software, termed a virtual router, enables bidirectional data exchange between the virtual controller and the DT. In contrast, if the physical controller is selected, the hardware router is required to establish the communication between the controller and the computer, thereby enabling bidirectional data exchange.

In conclusion, virtual commissioning and traditional commissioning differ primarily in their commissioning objects. Virtual commissioning utilizes virtual simulation models as the commissioning objects, while traditional commissioning uses physical entities. Beyond the distinction in commissioning objects, they also diverge in the commissioning environment, methods of controller implementation, and modes of interaction with the commissioning objects. These differences further result in their distinct advantages and limitations for virtual and traditional commissioning in application scenarios, commissioning processes, cost investments, and so on. Traditional commissioning, being a more familiar approach for electrical engineers, involves direct commissioning on the physical entity, yielding immediate commissioning results. However, precisely because traditional commissioning operates on physical entities, it is constrained by spatial and temporal limitations. Additionally, system failures during traditional commissioning directly impact physical equipment, risking economic losses and even safety hazards. In contrast, virtual commissioning requires only a computer running the virtual

commissioning object. Moreover, the simulation speed of the virtual object can be software-configured, and data acquisition methods and interactive operations can be designed with greater flexibility and diversity. This significantly overcomes spatial and temporal constraints, improves the commissioning efficiency, and ensures the safety of both equipment and commissioning personnel. However, realizing these advantages depends critically on high-fidelity DT models; results obtained by commissioning based on low-fidelity DT models may lack practical validity. Notably, as DT technologies mature and manufacturing digitization advances, high-fidelity DTs are increasingly applied to the design, construction, and operation of production lines, thereby expanding the application of virtual commissioning across a growing range of industrial application scenarios.

## 2.3 Control system design process

Previous analyses and discussions on DTs predominantly focused on the perspective of model developers, who possess expertise in DT development methods, applied technologies, and operational principles. Their studies emphasize building efficient DTs and optimizing their performance. In contrast, this paper adopts the perspective of DT end-users —specifically, electrical control engineers who may lack in-depth knowledge of DT development technologies and consequently face challenges in practical applications. Under such conditions, critical questions arise: How can engineers evaluate whether a DT meets their application requirements? What systematic workflow should be followed to design a DT-based electrical control system? As illustrated in Fig 2, the proposed DT-based electrical control system design workflow encompasses the following key components: DT maturity analysis, controller selection, control program design, establishment of the virtual commissioning platform, virtual commissioning, control system optimization, and so on [21].

The maturity level of the DT must first be evaluated to verify that it meets the design requirements of the control system. If the DT fails to meet the requirements, it should be refined or replaced with a DT that satisfies the maturity requirements.

The next step involves determining the controller type, specifically whether to adopt a physical or a virtual controller, along with identifying the brand and model of the physical controller or the simulation software for the virtual controller. Once the controller is selected, the corresponding programming software can be finalized, enabling the development of the control program based on the specified control requirements.

Prior to initiating the commissioning work, a virtual commissioning platform must be established to enable bidirectional real-time communication between the controller and the DT. Then, the control program is loaded into the controller, which drives the DT to execute production actions while the DT concurrently feeds back the operational status to the controller in real time. Subsequent iterative refinement of the control system is conducted based on commissioning results until compliance with production process design specifications is achieved. Leveraging the software nature of the DT, engineers can introduce disturbances and simulate fault scenarios. Consequently, the reliability of the control system can be comprehensively tested through the virtual commissioning process. Furthermore, by analyzing the operating parameters from the commissioning process based on DTs, potential flaws in the DT itself can be identified, facilitating iterative optimization of the DT [22,23].

## 3. Analysis of DT maturity

The quality of DTs is a critical factor influencing control system design. To explore the design methodologies of the DT-based electrical control system, it is necessary to first assess the maturity of the DT to evaluate its suitability for fulfilling the design requirements of the control system. Moreover, as application scenarios dictate distinct functional requirements, maturity indicators for DT vary accordingly. This section examines the specific maturity indicator for DTs in the application scenarios of DT-based control system design.

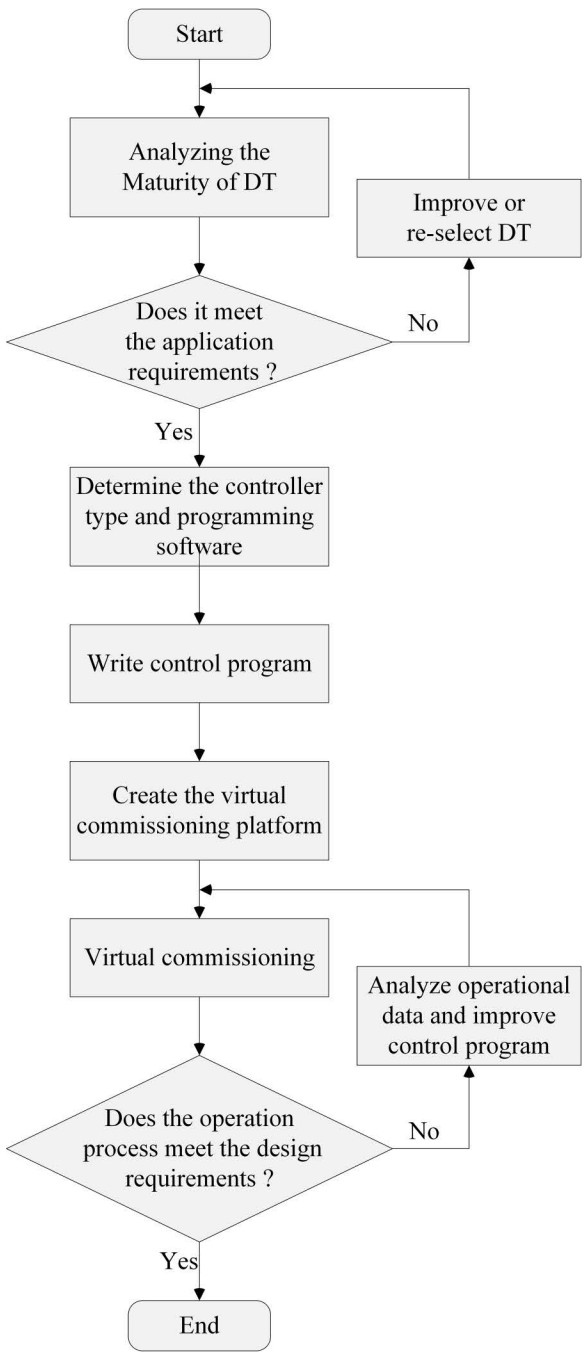

**Fig 2. Design process of DT-based electrical control system.**

### 3.1 DT maturity evaluation factors

The quality of DT is the decisive indicator for the DT-based control system design, as only a qualified DT provides a meaningful foundation for such engineering tasks. As shown in Fig 3, Tao Fei et al. [14,24] proposed a theory of DT maturity level, employing the five-dimension DT model encompassing 19 maturity evaluation factors as maturity evaluation

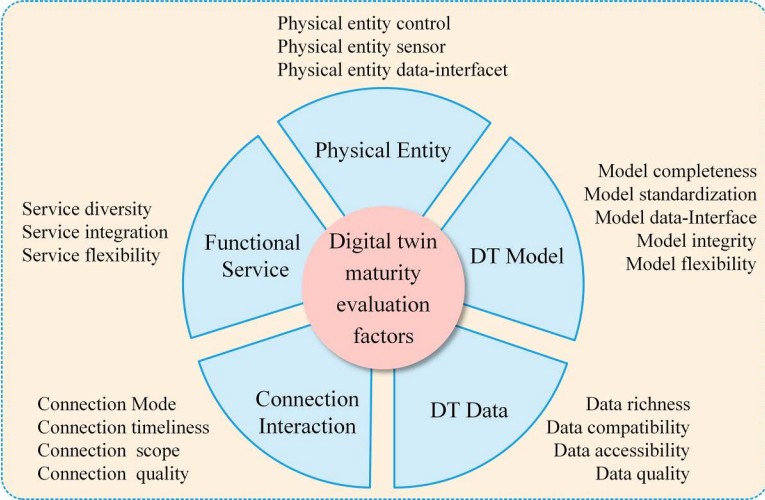

**Fig 3. DT maturity evaluation factors.**

criteria. From the perspective of electrical control engineers, this study further analyzes and defines the maturity indicators essential for DTs in control system design and summarizes the evaluation framework to assess DT maturity in this application scenario.

In the application scenarios of this study, the design of DT-based control systems is commonly used during the design phase of physical entities. This approach substitutes the physical entity controlled object with a DT for experimental validation, thereby enabling verification and optimization of the construction and operational plans for the physical entity [14]. During this process, physical entities are often under construction or have not yet been started, which eliminates the maturity requirements for the evaluation factors in the Physical Entities and Connection Interaction dimensions. Conversely, maturity requirements focus on the three dimensions: DT Model, DT Data, and Functional Service, as shown in Table 1.

**3.1.1 DT Model.** The DT Model is the primary dimension for directly evaluating the design maturity of the model itself, significantly influencing whether the application of control system design can be effectively implemented. Therefore, stringent requirements are imposed on the factors within this dimension. Specifically, the model must achieve high integrity to precisely replicate the physical structure and dynamic characteristics of the corresponding physical entity. Additionally, it must be equipped with standardized data interfaces for external communication to facilitate the I/O of dynamic and static parameters that meet the requirements of the target scenario. In the application scenarios of this study, the flexibility of DT models is not critical, as manual configuration should meet operational requirements.

**3.1.2 DT Data.** DT Data comprises design-phase data and operational data that reflect the operational status of DTs, with data diversity and quality as key indicators. As the diversity and accuracy of multidimensional data during the DT runtime increase, the reliability in achieving the design objectives of control systems is significantly enhanced. Data quality is a key indicator of this dimension, thus data generated by DTs should avoid duplication, omissions, ambiguities, and errors. The DT should integrate dynamic monitoring, verification, evaluation, and early warning mechanisms to detect and rectify data anomalies, ensuring the accuracy and reliability of data. Accessibility requires that users can easily retrieve and use DT data. The control system design does not impose stringent requirements on the data compatibility indicator, provided that it meets the data exchange protocols of the controller.

**3.1.3 Function Service.** The maturity of the Function Service dimension mainly affects the quality of virtual commissioning processes. The diversity indicator requires the capability of configuration, control, and visualization to the DT, as well as enabling data interaction with the controller. Because the key workflows of DT-based control system design

**Table 1. Maturity factors essential for control system design.**

| Dimension | Evaluation factor | Maturity requirement |
|---|---|---|
| DT Model | Model completeness | The system contains geometric, physical, motion, and rule models that accurately describe physical entities in multiple aspects, including geometry, internal structure, physical properties, and motion fidelity. |
| | Model standardization | Standardized data interfaces. |
| | Model data-Interface | Input and output(I/O) interfaces with dynamic and static parameters to meet current requirements. |
| | Model integrity | Effective integration of sub-models at the data, feature information, and model-based decision-making layer. |
| | Model flexibility | Enables manual configuration. |
| DT Data | Data diversity | Contains data from the design and manufacturing phases of the physical entity, as well as runtime data of the DT. |
| | Data compatibility | Supports standardized data exchange protocols with physical/virtual controller. |
| | Data accessibility | Provides access to data in the local databases and model files. |
| | Data quality | No duplication, omissions, ambiguities, or errors in raw data or data generated by functional services; complete mechanisms for dynamic monitoring, verification, evaluation, and early warning. |
| Functional Service | Service diversity | Enables configuration, control, visualization of DTs, and data interaction with controllers. |
| | Service integration | Most tightly coupled functional services during the operation and maintenance phases are integrated within a compatible software environment. |
| | Service flexibility | Supports manual configuration, matching, invocation, and optimization. |

are software-driven, stringent requirements are imposed on the integration indicator. Specifically, the service functions of the DT should be integrated within compatible software environments to facilitate real-time interaction with virtual controllers, virtual routers, and other simulation tools.

## 3.2 Application of DT maturity analysis

**3.2.1 DT structure and function.** The analysis process with the DT as the controlled object is fundamentally consistent with that of the physical entity. It requires an in-depth understanding of the DT's structure and function, workflow, and operational environment. The depth and breadth of this understanding determine whether the designed control system can optimally achieve its objectives.

The controlled object in this study is a DT of a six-floor elevator, implemented via the EET Basic simulation software developed by Beijing Dpro Technology Co., Ltd. The software presets a single-floor height of 3.1 meters, resulting in a total building height of approximately 22 meters. As shown in Fig 4, the interface comprises three main sections: a mode configuration panel (upper), an elevator operation visualization area (left), and a manual commissioning and status display area (right).

The elevator components, as shown in Fig 5, include: elevator car, traction motor, upper/lower limit switches, leveling sensors, external call buttons, and indicator lights on each floor. The elevator car is equipped with door control buttons, floor selection buttons, lighting, ventilation fans, safety light curtains, overload sensors, and so on.

The elevator's core functionality involves real-time responses to external call requests from various floors, controlling the traction motor to execute sequential motions such as startup, acceleration, constant-speed operation, deceleration, and braking. The elevator should properly respond to diverse passenger demands by effectively executing functionalities such as automatic leveling, door opening and closing, overload alerts, upper/lower limit protection, landing door interlock mechanisms, fault diagnosis and recovery protocols, and energy efficiency optimization capabilities.

**3.2.2 DT maturity analysis.** DT Model dimension: EET Basic software provides a high-fidelity simulation of a six-floor elevator, including its physical structure, operational environment, motion trajectories, monitoring/detection

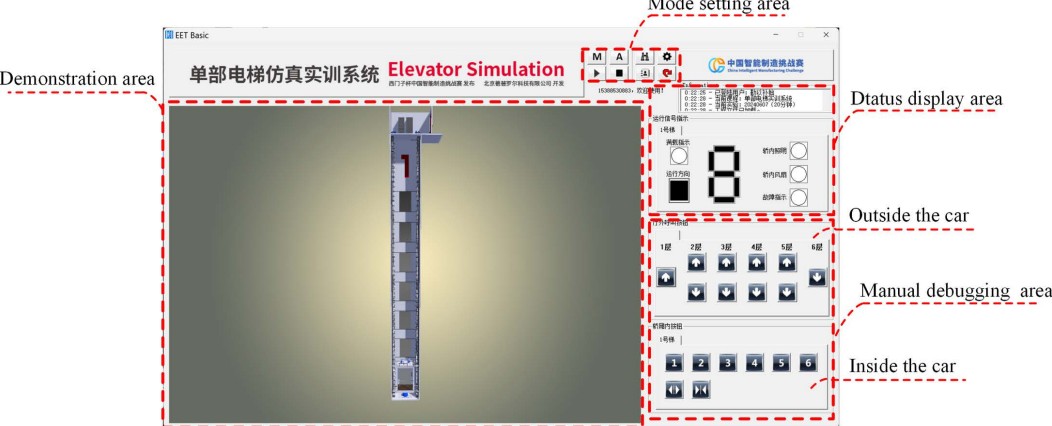

**Fig 4. EET Basic software interface.**

mechanisms, and 3D visualization model. The software integrates models and associated data for components such as floors, elevator cars, motors, and sensing devices. Standardized data interfaces enable real-time data exchange with the controller through a virtual router, simulating both normal operations and fault responses for the six-story elevator under controller commands. The software also supports user-configurable operating modes and environment parameters.

DT Data dimension: During runtime, the DT transmits real-time status data (e.g., call requests, display outputs, limit statuses, alarms) to the controller and simultaneously receives control signals (e.g., start, brake, stop, door control). When deviations occur in the control system, the DT will demonstrate the erroneous operational outcomes resulting from them. Furthermore, the elevator DT evaluates operational quality across elevator workflows, with generated data directly reflecting the design quality of the control system.

Functional Service dimension: The elevator DT allows user-configurable I/O parameters, operational modes, and environmental conditions, alongside visualized monitoring of the operational processes. The software integrates 3D modeling, motion drivers, data exchange protocols, and display configurations into a unified platform, demonstrating strong environment compatibility to meet virtual commissioning requirements.

In summary, the maturity levels of the six-floor elevator DT across the DT Model, DT Data, and Functional Service dimensions meet the requirements of the control system design. The elevator DT provides a reliable foundation for the control system design and commissioning.

## 4. Control system design

### 4.1 Controller and associated software

The design of the control system was implemented through various software tools, with the Siemens S7-1500 series Programmable Logic Controller (PLC) as the controller. The design process required both PLC programming software and simulation software.

TIA Portal, a fully integrated automation software developed by Siemens, combines hardware configuration, programming, and debugging functions. In this study, TIA Portal V18 was employed for hardware configuration and programming [25]. S7-PLCSIM Advanced V5.0, a dedicated Siemens PLC simulation software, enables the operation of a virtual S7-1500 PLC at a specified IP address [3]. It also integrates a virtual router to facilitate real-time Ethernet communication among the TIA Portal, the virtual PLC, and the DT [26].

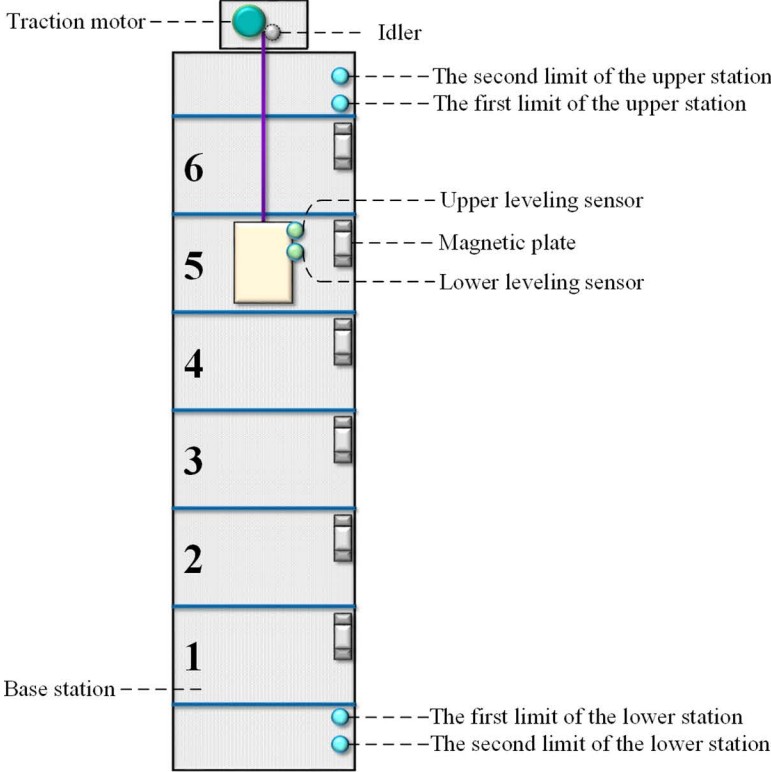

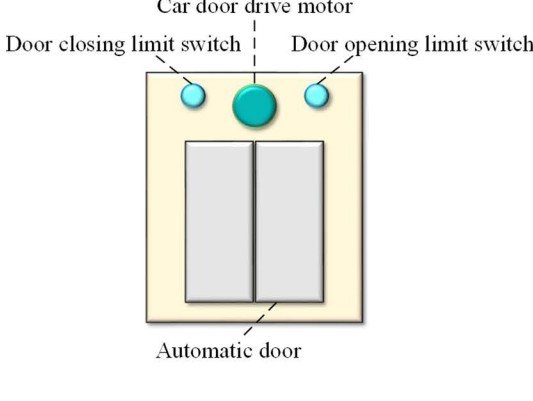

(A) Elevator shaft

(B) Elevator car

**Fig 5. Elevator composition diagram.**

## 4.2 Control program design

The overall flowchart of the control program is shown in Fig 6. Designed with a modular architecture, the program is partitioned into nine functional modules: initialization, floor number recording and display, external call signal processing and display, internal call signal processing and display, direction selection(up/down), motion control(upward/downward),

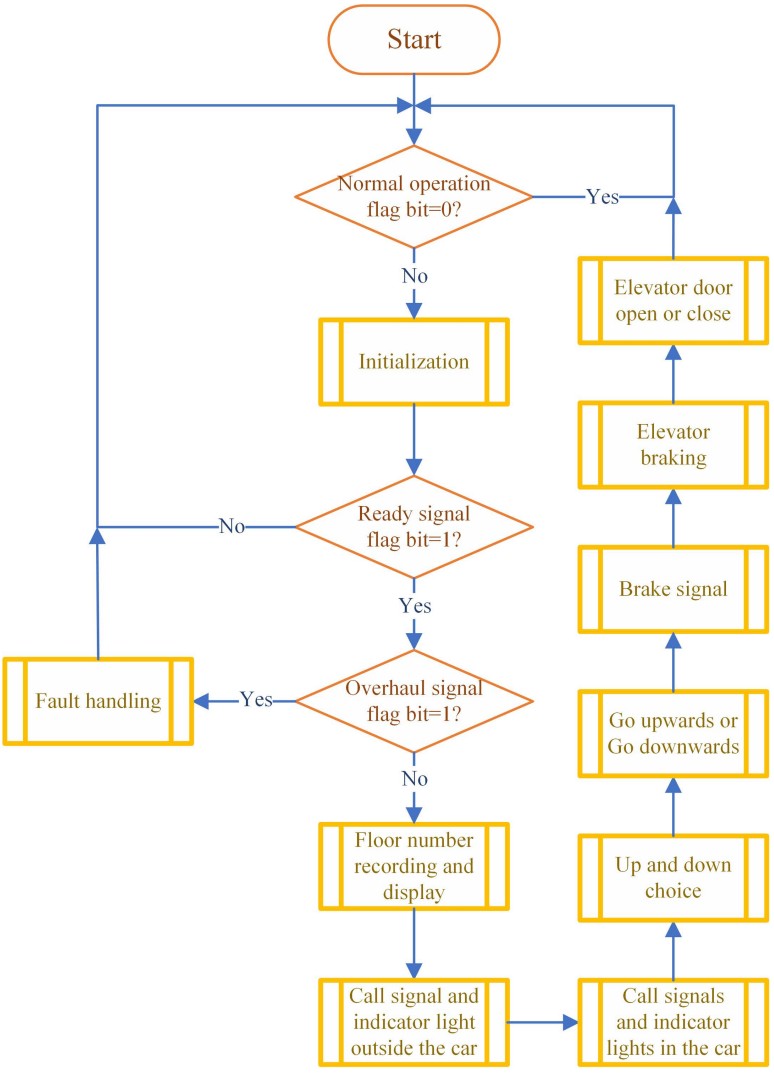

**Fig 6. The overall flowchart of the control program.**

braking and stop signal processing, leveling and stop, door operation, fault diagnosis and handling. These modules collectively implement the control logic for the 6-story elevator DT.

### 4.3 Virtual commissioning platform configuration

TIA Portal, S7-PLCSIM Advanced, and DT operate within the Windows environment. S7-PLCSIM Advanced integrates a virtual router, enabling real-time data exchange across the three platforms through compatible interfaces, thereby meeting virtual commissioning requirements. To validate the design quality of the control system, passenger behavior must be integrated into the elevator simulation. The EET Basic software simulates passenger call requests and generates fault signals by loading predefined project files, ensuring comprehensive verification of the control system performance and facilitating the system optimization.

**4.3.1 EET Basic software configuration.** The EET Basic software had to first be configured in experimental mode, after which it automatically loaded the predefined project files. It was necessary to manually configure the virtual PLC's IP address and the I/O data block address. The configuration parameters are shown in Fig 7.

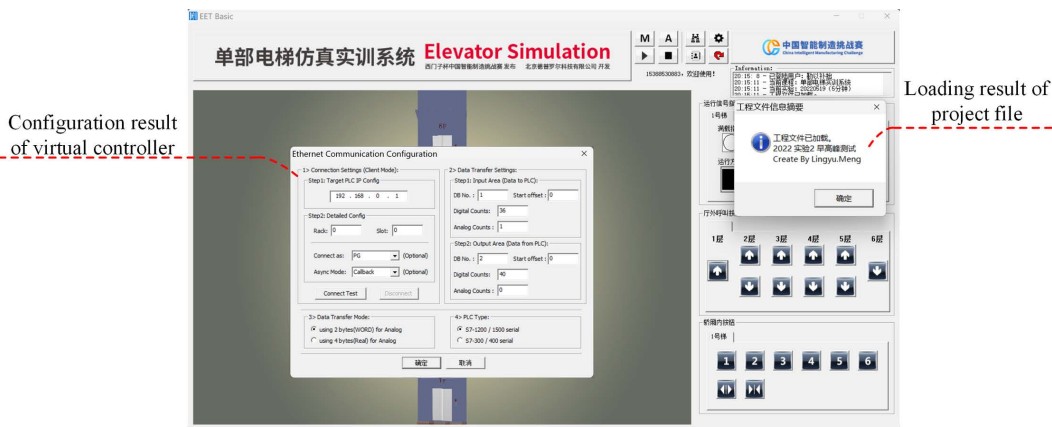

Configuration result of virtual controller

Loading result of project file

**Fig 7. EET Basic software configuration.**

**4.3.2 S7-PLCSIM Advanced software configuration.** The TCP/IP operation mode should first be selected to activate the virtual router function, enabling Ethernet communication between the virtual PLC and other platforms. The virtual PLC was initialized by specifying its name, IP address, and gateway, followed by clicking the start button. Successful establishment of the virtual PLC was confirmed when the PLC status indication (yellow for standby mode) and IP address appeared in the interface. The configuration parameters are shown in Fig 8.

**4.3.3 TIA Portal software configuration.** First, communication between TIA Portal and S7-PLCSIM Advanced was established by synchronizing the IP address. The control program and hardware configuration in TIA Portal were downloaded to the virtual PLC within the S7-PLCSIM Advanced. Finally, the virtual PLC was switched to online operational mode with the program monitoring function enabled. The configuration parameters are shown in Fig 9.

**4.3.4 Data exchange configuration.** The I/O data for the DT interactions with external platforms depend on the database established during the DT design phase. During the control system design, it was essential to establish address mapping between the I/O registers of the virtual PLC and the I/O registers of the DT prior to program design. To enhance the portability of program migration to the physical entity, two dedicated data blocks were created within the virtual PLC and mapped to the I/O data blocks of elevator DT, enabling bidirectional data exchange. This approach simplifies future parameter assignments between virtual and physical controllers by aligning DB block parameters with physical I/O points [27]. The configuration parameters are shown in Fig 10.

## 4.4 Virtual commissioning process

First, the automatic operation mode was selected in the EET Basic, after which the run button was clicked, triggering the elevator initialization according to preconfigured program settings. Upon completing initialization, the elevator entered standby mode. Passenger call signals were subsequently generated according to the predefined project file, and the elevator responded dynamically to these signals based on the control logic. The operational interface of EET Basic during runtime is shown in Fig 11. This interface provides real-time visualization of elevator motion, passenger call requests, and system status. Control program execution is simultaneously monitored via the TIA Portal. The simulation automatically terminates when reaching the preset duration specified in the project file, triggering the performance scoring interface. This evaluation module quantifies the elevator control system performance through these critical metrics: initialization reliability, passenger transport efficiency, core functionalities validation, operational stability and safety, and energy efficiency. The scoring interface is shown in Fig 12.

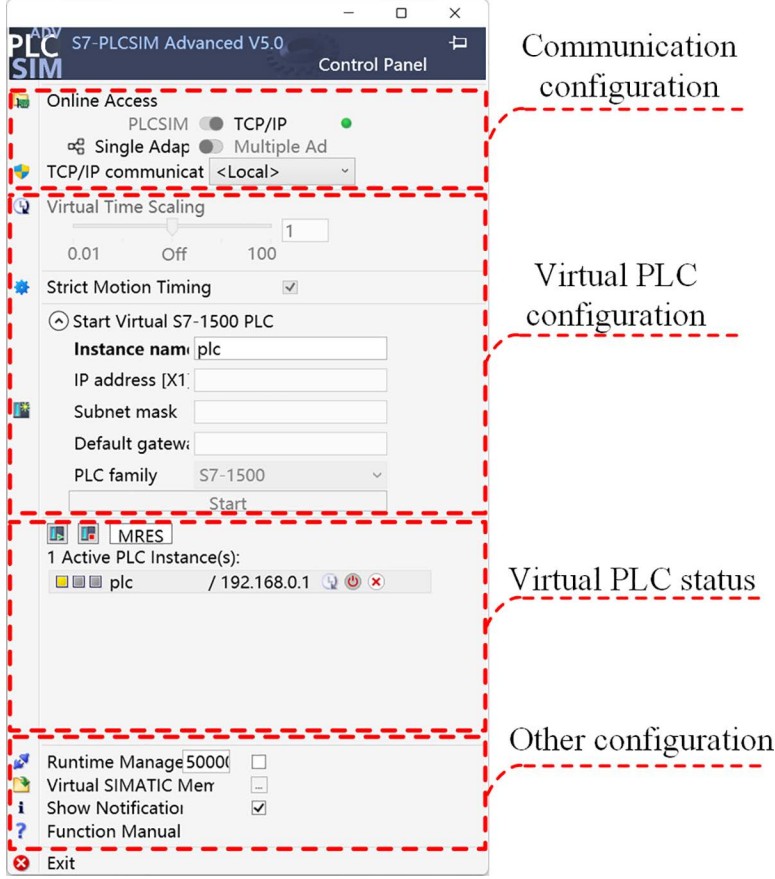

**Fig 8. S7-PLCSIM Advanced software configuration.**

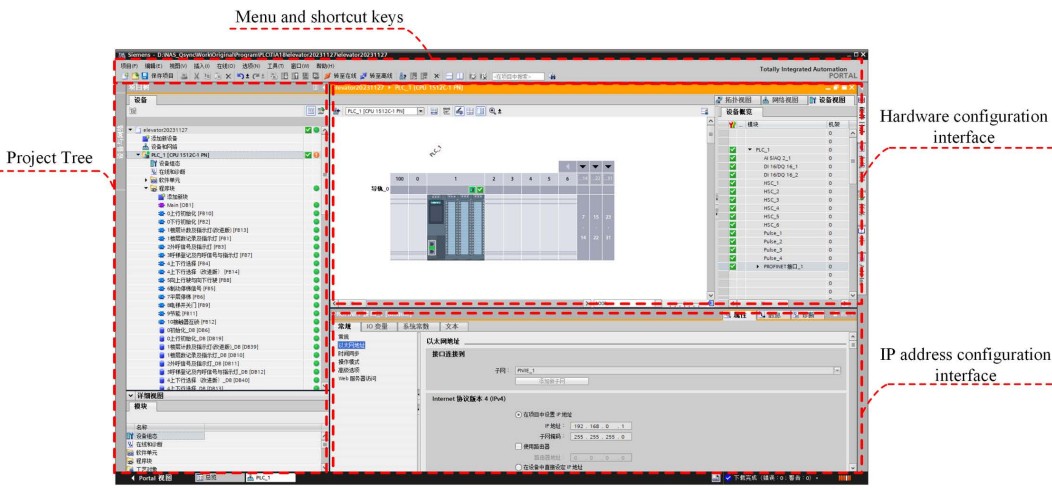

**Fig 9. TIA Portal software configuration.**

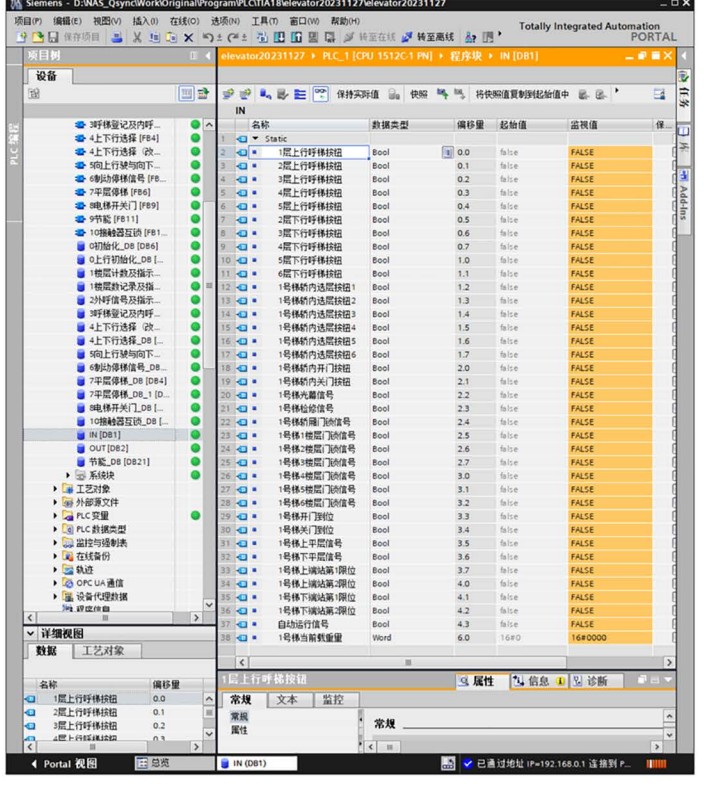
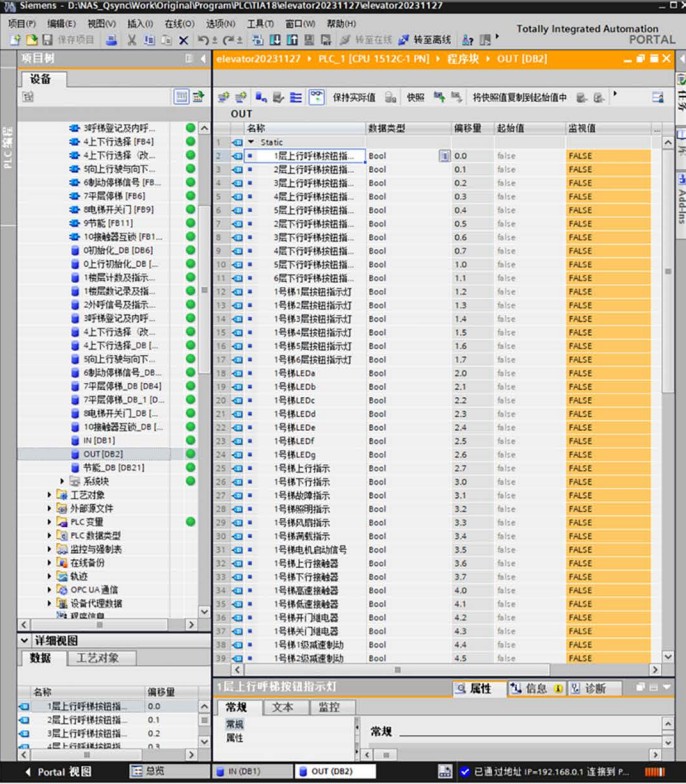

(A) Input data of virtual PLC

(B) Output data of virtual PLC

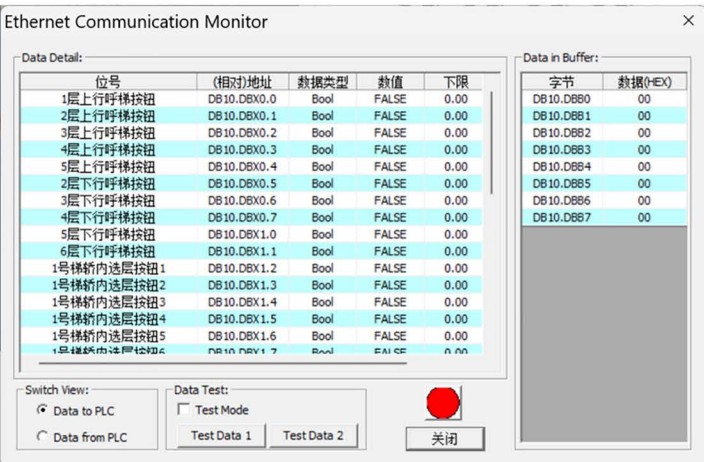
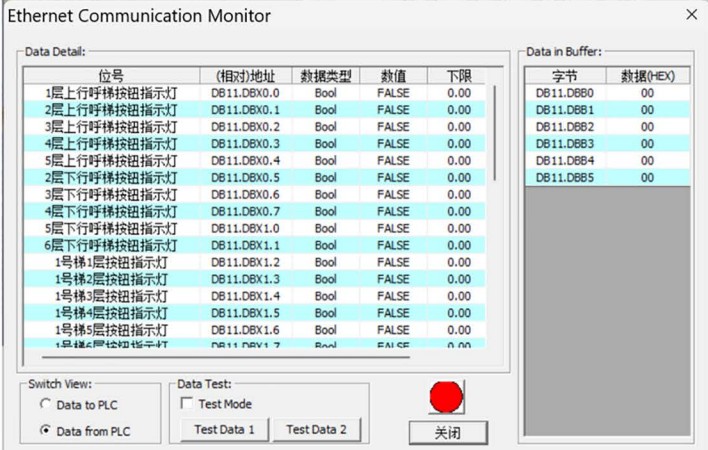

(C) Input data of DT

(D) Output data of DT

**Fig 10. Data exchange configuration.**

## 4.5 Result analysis and system improvement

Fig 13 shows the evaluation results from four iterations of virtual commissioning. The vertical axis represents the scores assigned by the elevator DT based on the virtual commissioning results, while the horizontal axis specifies the scored categories. The total score of each virtual commissioning is composed of seven components: elevator initialization, morning

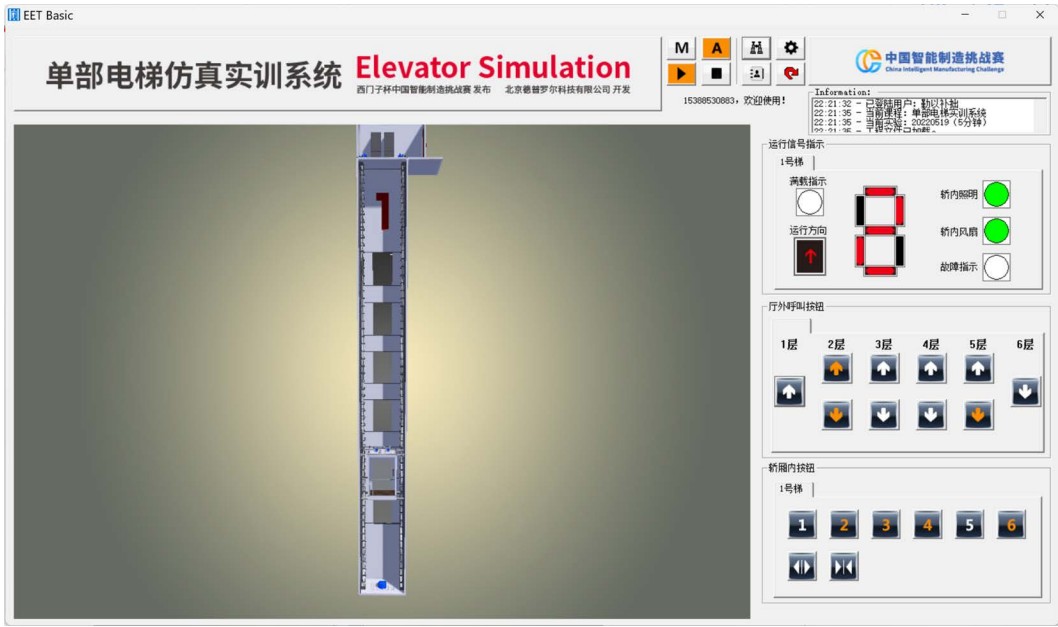

**Fig 11. Operational interface of EET Basic.**

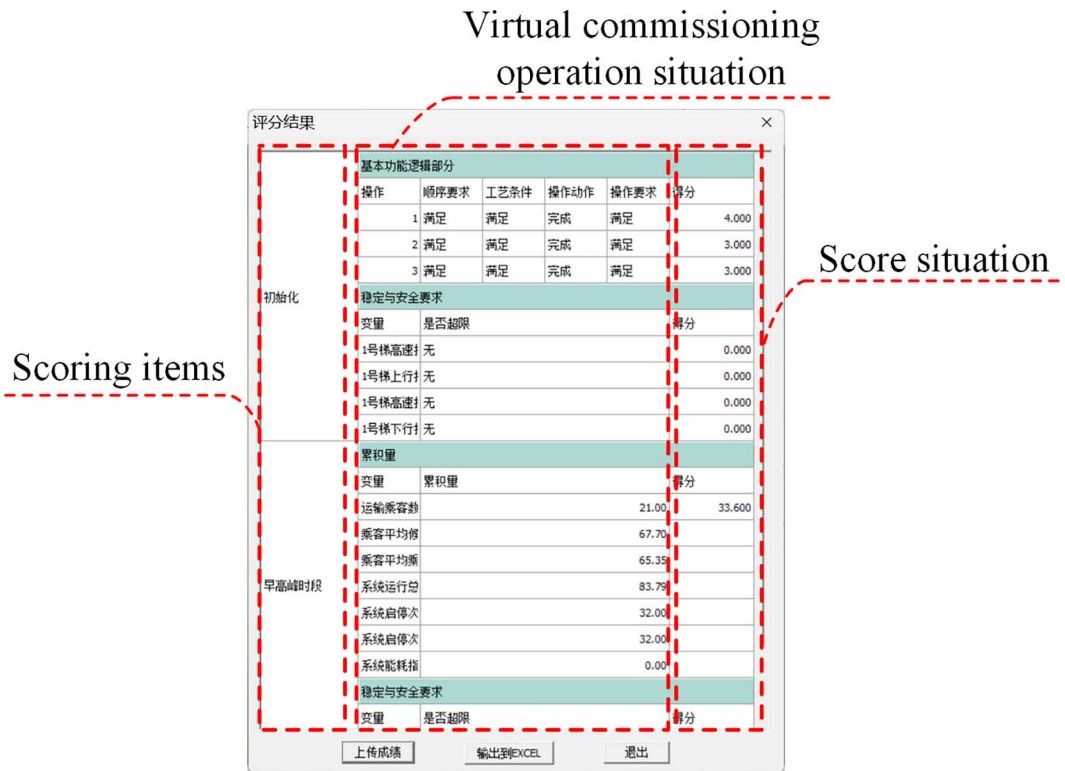

**Fig 12. Scoring interface of elevator virtual commissioning.**

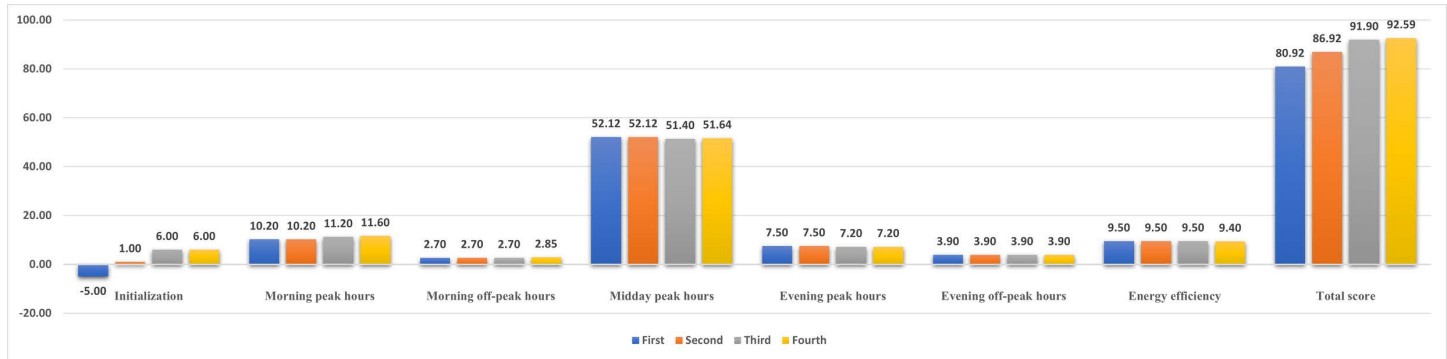

**Fig 13. Results of iterative commissioning.**

peak hours, morning off-peak hours, midday peak hours, evening peak hours, evening off-peak hours, and energy efficiency. Under the virtual PLC regulation, the scoring mechanism evaluates the elevator DT's ability to respond accurately and promptly to signals (e.g., initialization, call requests, stop) generated by the project file. Higher scores are awarded for efficient passenger delivery within specified timeframes and reduced energy consumption during operation/standby modes. In addition, the elevator DT assesses the control system's implementation of safety-critical functions, including electrical interlocking, positional limit protection, and fault handling mechanisms. The absence of these safety protection functions triggers a predefined score deduction.

During the first virtual commissioning trial, the elevator DT incurred a 5-point deduction due to two initialization-phase deficiencies: inaccurate parking position alignment and suboptimal switching between high-speed and low-speed contactors. The second trial rectified parking positioning (awarded 6 points) but retained the 5-point penalty for unresolved contactor switching issues, yielding a net initialization score of 1 point. The third trial successfully resolved all initialization defects, achieving a 6-point score. The fourth trial implemented motion control algorithm optimizations that enhanced the efficiency of transporting passengers during peak hours, culminating in a total score of 92.59 points, achieving near-ideal system performance.

In summary, the software-generated performance scores enable the identification of control system deficiencies during the runtime of the elevator DT and the prioritization of optimization. These metrics serve as the basis for implementing targeted control logic refinements. Following optimization, iterative re-commissioning is conducted until all metrics meet elevator operation requirements. In addition, throughout the commissioning process, operational anomalies and optimization requirements observed within the DT provide critical insights for enhancing the fidelity and functionality of the DT itself.

## 5. Conclusions

This paper investigates and practices the design method of the DT-based electrical control system from the perspective of electrical control engineers. Against the backdrop of the increasingly widespread application of DT-related technologies in manufacturing, we propose a design pathway that electrical control engineers can follow when designing the control system with DT as the controlled object. The main research contributions are summarized as follows:

### 5.1 Research summary

This paper systematically addresses the critical challenges in designing electrical control systems based on DT. First, the quality of the DT is the most critical factor. Prior to initiating the electrical control system design, engineers must ensure that the DT meets the application-specific requirements. Another key challenge lies in the inherent nature of DT as the software-based system deployed on the computational platform, so it is more suitable to use virtual commissioning for the

completion of the control system commissioning. The virtual commissioning with DT differs significantly from the traditional commissioning method with physical entities, which requires electrical engineers to develop expertise in virtual commissioning techniques. Furthermore, we propose the design process of the electrical control system based on DT. Electrical control engineers can apply this method when designing the control system based on DT.

Prior to commencing the detailed design of the control system, it is imperative to verify whether the DT meets the design requirements. The study analyzes the specific quality indicator for DT in electrical control system design and proposes a maturity evaluation framework comprising 12 influencing factors across three dimensions of the DT Model, DT Data, and Functional Service. Additionally, it specifies detailed technical specifications for each factor.

## 5.2 Practical verification

A case study of a six-floor elevator DT was conducted to evaluate its maturity, demonstrating compliance with the technical specifications required for designing the electrical control system. Applying the design workflow proposed above, we designed the electrical control system for the six-floor elevator DT. The virtual PLC was selected as the controller. The virtual commissioning platform was constructed by utilizing Siemens TIA Portal V18 and S7-PLCSIM Advanced V5.0 on the Windows operating system. The six-floor elevator control program was designed and then validated through virtual commissioning. Thus, the design of the electrical control system for the six-floor elevator DT was completed.

Experimental results demonstrate that the DT model that complies with maturity indicators can serve as the controlled object to design the electrical control system and validate their design quality through iterative virtual commissioning. By controlling the DT via the controller, engineers can validate the feasibility and reliability of the mechanical structure, motion trajectory, production workflow, and fault-handling mechanism during the design and construction phases of the corresponding physical entity. This approach significantly contributes to shortening the production line development cycle, reducing the physical testing cost, enhancing the commissioning efficiency, ensuring the operation quality, and refining the fault management mechanism.

## 5.3 Challenges and limitations

Although the DT-based electrical control system design method explored in this study enables electrical control engineers (who lack in-depth familiarity with DT design principles) to efficiently utilize DTs for system design, commissioning, operation, and optimization, the implementation of this approach imposes specific requirements on both the DT and its operational environment.

First, the DT models of production line equipment must be already established, with their maturity aligned to the requirements of the electrical control system design application scenario. In addition, establishing a virtual commissioning environment is a systematic task that necessitates collaboration among components, including the DT, virtual controllers, virtual routers, and programming/commissioning tools. Underperformance in any component may compromise the entire process. Finally, the lack of standardization and generalizability in DT technologies limits the method's broader applicability. Application rules and commissioning environments for DTs may vary significantly across different projects, which not only increases engineers' learning costs but also diminishes theoretically achievable efficiency gains of the proposed method.

## 5.4 Future prospects

As digitalization in manufacturing advances and digital technologies mature, DT technology will assume an increasingly critical role in driving industrial innovation. DT will be deeply integrated with artificial intelligence, big data analytics, cloud computing, and other technologies to achieve a higher level of intelligence and automation. These factors will propel manufacturing toward greater flexibility, efficiency, and intelligence, presenting new challenges and opportunities for electrical control engineers. To support the high-speed, high-quality development of the industry, electrical engineers need to

constantly update their knowledge systems and engage in the exploration and practical application of design methods of the control system based on DT.

## Supporting information

**S1 File. Program and scoring records.**
(ZIP)

## Author contributions

**Conceptualization:** Hao Shi.

**Project administration:** Qingliang Zhu.

**Software:** Hao Shi.

**Validation:** Jiayang Shi, Tao Zhao.

**Writing – original draft:** Hao Shi.

**Writing – review & editing:** Hao Shi, Jingde Bo.

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
