## [Decision Letter · Decision Letter 0]

Dear Dr. Shi,

Thank you for submitting your manuscript to PLOS ONE. After careful consideration, we feel that it has merit but does not fully meet PLOS ONE’s publication criteria as it currently stands. Therefore, we invite you to submit a revised version of the manuscript that addresses the points raised during the review process.

ACADEMIC EDITOR: Major revisions

We look forward to receiving your revised manuscript.

Kind regards,

Agbotiname Lucky Imoize

Academic Editor

PLOS ONE

Journal Requirements:

Please confirm at this time whether or not your submission contains all raw data required to replicate the results of your study. Authors must share the “minimal data set” for their submission. PLOS defines the minimal data set to consist of the data required to replicate all study findings reported in the article, as well as related metadata and methods (https://journals.plos.org/plosone/s/data-availability#loc-minimal-data-set-definition ).

If your submission does not contain these data, please either upload them as Supporting Information files or deposit them to a stable, public repository and provide us with the relevant URLs, DOIs, or accession numbers. For a list of recommended repositories, please see https://journals.plos.org/plosone/s/recommended-repositories .

Additional Editor Comments:

Dear Authors,

Kindly revise the manuscript according to the reviewers comments.

Reviewers' comments:

Reviewer's Responses to Questions

**Comments to the Author**

1. Is the manuscript technically sound, and do the data support the conclusions?

Reviewer #1: Yes

Reviewer #2: Yes

2. Has the statistical analysis been performed appropriately and rigorously?

Reviewer #1: Yes

Reviewer #2: Yes

3. Have the authors made all data underlying the findings in their manuscript fully available?

Reviewer #1: No

Reviewer #2: Yes

4. Is the manuscript presented in an intelligible fashion and written in standard English?

Reviewer #1: Yes

Reviewer #2: Yes

Reviewer #1: This paper presents a PLC control system design method based on digital twin. However, there are some issues that need to be addressed as follows.

1) Section 0: why should digital twins be integrated into electrical control systems? The challenges to be addressed by this work should be clarified.

2) Section 1.2: the differences between virtual commissioning and traditional commissioning methods are not summarized clearly. What are the advantages and disadvantages of virtual commissioning and traditional commissioning?

3) Section 1.3: the control system design process is a reference to someone else's work and the innovation of this work should be clarified.

4) Section 3.3: the designed software interface should clearly identify the different areas and describe their functions in English, rather than merely presenting a screenshot of the interface.

5) Section 3.5: the vertical axis in Figure 13 lacks a clear explanation. The meaning of this coordinate should be clarified, and the software results should be quantitatively presented.

6) Section 4: the methodology employed in this work is not adequately addressed, and the experimental results are not quantified. These aspects should be explicitly included to provide a more comprehensive conclusion.

The manuscript in its current shape is not ready for publication in the Journal. The issues mentioned above must be addressed and a revised manuscript be resubmitted for review.

Reviewer #2: 1.The number of references cited in this manuscript is relatively limited, which to some extent constrains the theoretical depth and breadth of the article. It is recommended that the author broaden the scope of literature searches in subsequent research and writing, incorporating citations of classic literature and the latest research findings in the relevant field.

2.I have observed instances of nearly identical paragraphs. Specifically, within the 2.2 Application of DT maturity analysis section, the paragraphs titled "DT Model dimension" and "Dimension of DT model," as well as the opening paragraphs of the "In summary" section, exhibit a high degree of similarity, seemingly reiterating the same information. Avoiding content repetition is crucial in academic writing, and it is advisable for the author to revise and adjust these paragraphs accordingly.

3.The conclusion analysis constitutes a vital component of the article, yet the conclusion analysis and system improvement section (3.5 Result analysis and system improvement) appears somewhat underdeveloped, failing to fully explore the significance and potential implications of the research findings. For instance, based on the scoring results, what specific issues exist within the control system during elevator operation, and what corresponding optimization methods are proposed?

4.The description of Fig. 13 lacks clarity, and there are missing annotations on the graphic, such as the units for the numerical values on the vertical axis, which may confuse readers. It is recommended that the author provide a more detailed description of the chart, including explanations of the horizontal and vertical axes, as well as the legends.

**Do you want your identity to be public for this peer review?** For information about this choice, including consent withdrawal, please see our Privacy Policy

Reviewer #1: No

Reviewer #2: No

---

## [Author Response · Author response to Decision Letter 1]

21 Jan 2025

Dear Reviewers,

Thank you very much for your comments and professional advice. These comments are of great help in enhancing the academic rigor of our manuscript, and they also have guiding significance for us to carry out relevant research work and write academic papers in the future. We would like to express our sincere gratitude again! According to your advice and requirements, we have revised and improved the manuscript. We added the missing content and necessary explanations, conducted quantitative analysis of the experimental data, improved annotations of several figures, and deleted unnecessary paragraphs. We greatly appreciate your re-review of the revised manuscript. We look forward to further improvements of our work. The following are the specific details of the manuscript revisions based on the reviewers' comments:

Reviewer #1:

Comment 1: Section 0: why should digital twins be integrated into electrical control systems? The challenges to be addressed by this work should be clarified.

Response:

We are extremely grateful to you for pointing out the issue. The previous manuscript did not clearly explain this issue. Now, we have added relevant content to the introduction section, clarifying that a complete Digital Twin is not merely the Digital Twin of production equipment, but should also include the Digital Twin of the corresponding electrical control system. Moreover, we have explained the important role of the Digital Twin of the electrical control system in the design, commissioning, and process improvement of the production line.

Comment 2: Section 1.2: the differences between virtual commissioning and traditional commissioning methods are not summarized clearly. What are the advantages and disadvantages of virtual commissioning and traditional commissioning?

Response:

We are extremely grateful to you for pointing out the issue. The previous manuscript did not have a clear summary of the differences between virtual commissioning and traditional commissioning methods, and the respective advantages and disadvantages of these two methods were not expounded clearly. Now, we have added relevant content to clearly summarize the differences between them and their respective advantages and disadvantages.

Comment 3: Section 1.3: the control system design process is a reference to someone else's work and the innovation of this work should be clarified.

Response:

We are extremely grateful to you for pointing out the issue. The previous manuscript did not clearly summarize the innovativeness of this research work. Now, we have added relevant content to clarify that the uniqueness of this research lies in the following: From the perspective of the end-user of Digital Twin, for an electrical control engineer who doesn't have in - depth knowledge of the relevant technologies for establishing Digital Twin, what kind of path should be followed to achieve the design of an electrical control system based on Digital Twin.

Comment 4: Section 3.3: the designed software interface should clearly identify the different areas and describe their functions in English, rather than merely presenting a screenshot of the interface.

Response:

We are extremely grateful to you for pointing out the issues with the figures in Section 3.3. We have revised the relevant figures.

The interface related to Figure 7 has already appeared in Figure 4 earlier. Therefore, we have annotated the functions of each part of the software interface in Figure 4. Meanwhile, we have clearly annotated the significance of the newly added content in Figure 7.

In Figure 8, we add annotations for the relevant content.

To be more in line with the description of the relevant content, we have replaced Figure 9 and added annotations.

Also, to better match the relevant description, we have revised Figure 10 by adding two figures. The comparison between the figures can more clearly demonstrate how the data exchange settings are implemented.

In Figure 12, we add annotations for the relevant content.

Comment 5: Section 3.5: the vertical axis in Figure 13 lacks a clear explanation. The meaning of this coordinate should be clarified, and the software results should be quantitatively presented.

Response:

We are extremely grateful to you for pointing out the issue. When referring to Figure 13 in the manuscript, we have added relevant content. Specifically, we have explained the meanings of the ordinate and abscissa, changed the display method to more prominently highlight the scores of each item. Moreover, we have introduced the main influencing factors of the scores in the four virtual commissioning processes, as well as the improvements made to the virtual control system based on the feedback results. This clarifies the significance of the presented data.

Comment 6: Section 4: the methodology employed in this work is not adequately addressed, and the experimental results are not quantified. These aspects should be explicitly included to provide a more comprehensive conclusion.

Response:

We are extremely grateful to you for pointing out the issue. We have added relevant content in Sections 3.5 and 4. First, in Section 3.5, we added explanations and analyses of the experimental data. Then, in Section 4, we added the summary and elaboration of the research work and methods involved in the manuscript, hoping to provide a more comprehensive conclusion.

Reviewer #2:

Comment 1: The number of references cited in this manuscript is relatively limited, which to some extent constrains the theoretical depth and breadth of the article. It is recommended that the author broaden the scope of literature searches in subsequent research and writing, incorporating citations of classic literature and the latest research findings in the relevant field.

Response:

We sincerely appreciate your valuable suggestions. As electrical control engineers mainly engaged in technical application work, our exploration and research on the depth and breadth of the theory of our article are indeed lacking. We will definitely strengthen our efforts in this regard during our subsequent research and writing. We will search for and study more classic literatures and the latest research findings. We believe these efforts will be of great help in improving our work capabilities and research levels. Thank you again for your precious suggestions!

Comment 2: I have observed instances of nearly identical paragraphs. Specifically, within the 2.2 Application of DT maturity analysis section, the paragraphs titled "DT Model dimension" and "Dimension of DT model," as well as the opening paragraphs of the "In summary" section, exhibit a high degree of similarity, seemingly reiterating the same information. Avoiding content repetition is crucial in academic writing, and it is advisable for the author to revise and adjust these paragraphs accordingly.

Response:

We are extremely grateful to you for pointing out the issue. It was indeed due to our oversight that these repetitive contents occurred. Your review has helped us avoid such elementary mistakes. We have deleted the repetitive contents. Thank you again!

Comment 3: The conclusion analysis constitutes a vital component of the article, yet the conclusion analysis and system improvement section (3.5 Result analysis and system improvement) appears somewhat underdeveloped, failing to fully explore the significance and potential implications of the research findings. For instance, based on the scoring results, what specific issues exist within the control system during elevator operation, and what corresponding optimization methods are proposed?

We are extremely grateful to you for pointing out the issue. Regarding this matter, when referring to Figure 13 in the manuscript, we specifically introduced the main influencing factors of the scores in the four virtual commissioning processes, as well as the improvements made to the virtual control system based on the feedback results. This demonstrates the assistance of the virtual commissioning process in enhancing the virtual control system.

Comment 4: The description of Fig. 13 lacks clarity, and there are missing annotations on the graphic, such as the units for the numerical values on the vertical axis, which may confuse readers. It is recommended that the author provide a more detailed description of the chart, including explanations of the horizontal and vertical axes, as well as the legends.

Response:

We are extremely grateful to you for pointing out the issue. When referring to Figure 13 in the manuscript, we have added relevant content. Specifically, we have explained the meanings of the ordinate and abscissa, changed the display method to more prominently highlight the scores of each item. Moreover, we have introduced the main influencing factors of the scores in the four virtual commissioning processes, as well as the improvements made to the virtual control system based on the feedback results. This clarifies the significance of the presented data.

Thank you very much for your attention and time. Look forward to hearing from you.

Yours sincerely

Hao Shi

Corresponding Author

Department of Electrical Engineering and Automation Shanxi Institute of Technology Yangquan Shanxi, China

---

## [Decision Letter · Decision Letter 1]

Dear Dr. Shi,

Thank you for submitting your manuscript to PLOS ONE. After careful consideration, we feel that it has merit but does not fully meet PLOS ONE’s publication criteria as it currently stands. Therefore, we invite you to submit a revised version of the manuscript that addresses the points raised during the review process.

**ACADEMIC EDITOR: Minor revisions**

We look forward to receiving your revised manuscript.

Kind regards,

Agbotiname Lucky Imoize

Academic Editor

PLOS ONE

Journal Requirements:

Additional Editor Comments:

Dear Authors,

Please revise the paper according to the reviewers' recommendations and improve the English usage significantly.

Thank you.

Reviewers' comments:

Reviewer's Responses to Questions

**Comments to the Author**

Reviewer #1: All comments have been addressed

Reviewer #2: All comments have been addressed

Reviewer #3: All comments have been addressed

2. Is the manuscript technically sound, and do the data support the conclusions?

Reviewer #1: Yes

Reviewer #2: Yes

Reviewer #3: Yes

3. Has the statistical analysis been performed appropriately and rigorously?

Reviewer #1: Yes

Reviewer #2: Yes

Reviewer #3: Yes

4. Have the authors made all data underlying the findings in their manuscript fully available?

Reviewer #1: Yes

Reviewer #2: Yes

Reviewer #3: Yes

5. Is the manuscript presented in an intelligible fashion and written in standard English?

Reviewer #1: Yes

Reviewer #2: No

Reviewer #3: Yes

Reviewer #1: The comments have been addressed in the revised manuscript. It is suggested to be accepted for publication.

Reviewer #2: 1.The abstract does not clearly state the innovation of this study. It is suggested to supplement the explanation of the innovative aspects of this study.

2.The language is repetitive and verbose. It is suggested to avoid absolute and colloquial expressions, unify the tenses, and make the language more concise.

3.The conclusion section mixes the research summary, practical verification, and future prospects in a way that lacks clear hierarchical structure. It is recommended to clearly separate these three parts and add a description of the limitations for the future, which will facilitate the reading and research of the readers.

Reviewer #3: The manuscript is technically sound. The methodology for designing and verifying PLC control systems using a Digital Twin is logical and well-detailed. The iterative approach to virtual commissioning and the analysis of commissioning results effectively demonstrate the reliability and efficiency improvements enabled by the DT approach. The data derived from multiple commissioning stages supports the conclusion that the DT-based design method improves efficiency and reduces errors. The statistical analysis is appropriate for the type of research presented. The iterative virtual commissioning process and the associated scoring system provide a quantitative evaluation of control system performance. The explanation of the scoring metrics and the detailed breakdown of improvements across commissioning iterations contribute to the rigor of the evaluation. The authors have declared that all relevant data are included within the manuscript and its supporting information files. The manuscript provides detailed descriptions of the virtual commissioning results, including scores and performance evaluations. the manuscript is generally well-written and presented in standard English. The structure is logical, with a clear progression from problem identification to methodology, results, and conclusions. However, there are occasional grammatical errors and awkward phrasings that could benefit from minor editorial improvements.

Comments to the Author:

1. The manuscript addresses a highly relevant and current topic in industrial automation and digital manufacturing.

2. The integration of Digital Twin technology with PLC control system design is clearly explained and demonstrated through a practical application.

3. The iterative virtual commissioning approach is well-documented and showcases the value of DTs in optimizing control systems before physical implementation.

4. The detailed explanation of the DT maturity analysis and the evaluation criteria provides a robust framework that could be beneficial to other researchers and practitioners.

5. The figures and illustrations significantly aid the reader’s understanding of the methodology and results.

6. The manuscript effectively highlights the advantages of virtual commissioning, such as reducing testing costs and improving system reliability, which are critical considerations in modern manufacturing environments.

Suggested Minor Revisions:

1. Review the manuscript for minor grammatical and typographical issues to improve readability.

2. Ensure consistency in terminology, particularly regarding “virtual commissioning,” “Digital Twin,” and related terms.

3. Consider expanding the discussion on potential limitations or challenges associated with implementing DT-based control system design in different industrial contexts.

**Do you want your identity to be public for this peer review?** For information about this choice, including consent withdrawal, please see our Privacy Policy

Reviewer #1: No

Reviewer #2: No

Reviewer #3: **Yes: ** Richard Govada Joshua

---

## [Author Response · Author response to Decision Letter 2]

5 May 2025

Reviewer #1:

Comment: The comments have been addressed in the revised manuscript. It is suggested to be accepted for publication.

Response:

We sincerely appreciate your thorough re-review of the manuscript. It is both an honor and a privilege to receive your recognition of our work.

Reviewer #2:

Comment 1: The abstract does not clearly state the innovation of this study. It is suggested to supplement the explanation of the innovative aspects of this study.

Response:

We sincerely appreciate your valuable suggestions. We have highlighted the innovation of this study in the abstract. Specifically, we emphasize that this study explores a method for designing Digital Twin (DT)-based electrical control systems from the perspective of an electrical control engineer with limited understanding of DT design and development technologies. This approach addresses the implementation challenges faced by practitioners with limited knowledge of DT technologies. Thank you again for your precious suggestions!

Comment 2: The language is repetitive and verbose. It is suggested to avoid absolute and colloquial expressions, unify the tenses, and make the language more concise.

Response:

We are extremely grateful to you for pointing out the issue. Due to our lack of English writing experience, the previous manuscript had issues in language and tenses. Your review has helped us avoid such basic errors. To address these, we reviewed the publisher’s guidelines and extensively analyzed published academic literature to enhance our understanding of tense conventions and language norms. The revisions were guided by the following principles:

Tense: Simple present tense is applied to describe the research background, objectives, and discussion of results. Simple past tense is used to detail experimental procedures and outcomes.

Language: Avoidance of absolute or overly assertive statements. Simplification of verbose sentences and correction of grammatical errors. Replacement of informal expressions with formal terminology.

We have thoroughly revised the entire manuscript, including the title, to enhance its readability and compliance with academic standards. Detailed revisions are highlighted in the “Revised Manuscript with Track Changes” file. The text highlighted in red indicates modifications or additions, while the text highlighted in red with strikethroughs indicates the deleted parts. Thank you again for your review!

Comment 3: The conclusion section mixes the research summary, practical verification, and future prospects in a way that lacks clear hierarchical structure. It is recommended to clearly separate these three parts and add a description of the limitations for the future, which will facilitate the reading and research of the readers.

Response:

We are extremely grateful to you for pointing out the issue. We have restructured the conclusion section into three subsections: Research Summary, Practical Validation, and Future Prospects. Additionally, we added a dedicated discussion on the potential limitations of the research content when applied in future contexts. These revisions are intended to facilitate the reading and research of the readers.

Reviewer #3:

Comment: However, there are occasional grammatical errors and awkward phrasings that could benefit from minor editorial improvements.

Response:

We are extremely grateful to you for pointing out the issue. Due to our lack of English writing experience, the previous manuscript had issues in language and tenses. Your review has helped us avoid such basic errors. To address these, we reviewed the publisher’s guidelines and extensively analyzed published academic literature to enhance our understanding of tense conventions and language norms. The revisions were guided by the following principles:

Tense: Simple present tense is applied to describe the research background, objectives, and discussion of results. Simple past tense is used to detail experimental procedures and outcomes.

Language: Avoidance of absolute or overly assertive statements. Simplification of verbose sentences and correction of grammatical errors. Replacement of informal expressions with formal terminology.

We have thoroughly revised the entire manuscript, including the title, to enhance its readability and compliance with academic standards. Detailed revisions are highlighted in the “Revised Manuscript with Track Changes” file. The text highlighted in red indicates modifications or additions, while the text highlighted in red with strikethroughs indicates the deleted parts. Thank you again for your review!

Suggested Minor Revisions:

Comment 1: Review the manuscript for minor grammatical and typographical issues to improve readability.

Response:

We are extremely grateful to you for pointing out the issue. Due to our lack of English writing experience, the previous manuscript had issues in language and tenses. Your review has helped us avoid such basic errors. To address these, we reviewed the publisher’s guidelines and extensively analyzed published academic literature to enhance our understanding of tense conventions and language norms. The revisions were guided by the following principles:

Tense: Simple present tense is applied to describe the research background, objectives, and discussion of results. Simple past tense is used to detail experimental procedures and outcomes.

Language: Avoidance of absolute or overly assertive statements. Simplification of verbose sentences and correction of grammatical errors. Replacement of informal expressions with formal terminology.

We have thoroughly revised the entire manuscript, including the title, to enhance its readability and compliance with academic standards. Detailed revisions are highlighted in the “Revised Manuscript with Track Changes” file. The text highlighted in red indicates modifications or additions, while the text highlighted in red with strikethroughs indicates the deleted parts. Thank you again for your review!

Comment 2: Ensure consistency in terminology, particularly regarding “virtual commissioning,” “Digital Twin,” and related terms.

Response:

We are extremely grateful to you for pointing out the issue. The previous manuscript did have problems with inconsistent terminology. Your review helped us avoid these elementary mistakes. We have meticulously reviewed the entire manuscript ourselves to ensure strict consistency in key terms such as "virtual commissioning" and "Digital Twin". Thank you again for your review!

Comment 3: Consider expanding the discussion on potential limitations or challenges associated with implementing DT-based control system design in different industrial contexts.

Response:

We are extremely grateful to you for pointing out the issue. We have restructured the conclusion section into three subsections: Research Summary, Practical Validation, and Future Prospects. Additionally, we added a dedicated discussion on the potential limitations of the research content when applied in future contexts. These revisions are intended to facilitate the reading and research of the readers.

Thank you very much for your attention and time. Look forward to hearing from you.

Yours sincerely

Hao Shi

Corresponding Author

Department of Electrical Engineering and Automation Shanxi Institute of Technology Yangquan Shanxi, China

---

## [Editor Report · Decision Letter 2]

A generic method for PLC control system design based on Digital Twin

PONE-D-24-49574R2

Dear Dr. Shi,

We’re pleased to inform you that your manuscript has been judged scientifically suitable for publication and will be formally accepted for publication once it meets all outstanding technical requirements.

Kind regards,

Agbotiname Lucky Imoize

Academic Editor

PLOS ONE

Additional Editor Comments (optional):

Accept in current form.
---

## [Editor Report · Acceptance letter]

PONE-D-24-49574R2

PLOS ONE

Dear Dr. Shi,

I'm pleased to inform you that your manuscript has been deemed suitable for publication in PLOS ONE. Congratulations! Your manuscript is now being handed over to our production team.

Kind regards,

on behalf of

Mr. Agbotiname Lucky Imoize

Academic Editor

PLOS ONE